

# Finding a relationship between physicochemical characteristics and ionic composition of River Nworie, Imo State, Nigeria

Evelyn Ngozi Verla[1,*], Andrew Wirnkor Verla[2] and Christian Ebere Enyoh[2,*]

[1] Department of Environmental Technology, School of Environmental Science, Federal University of Technology, Owerri, Imo State, Nigeria, Owerri, Nigeria
[2] Group Research in Analytical Chemistry, Environment and Climate Change (GRACE&CC), Department of Chemistry, Imo State University, Owerri, Owerri, Nigeria
[*] These authors contributed equally to this work.

Corresponding author
Christian Ebere Enyoh,
cenyoh@gmail.com

## ABSTRACT

Water has been described as a universal solvent, and this is perhaps the strength behind its many uses. Despite this unique property, anthropogenic activities along its course and natural factors often determine the composition of water. In the current research, the portion of River Nworie having past Owerri town was sampled in the dry season 2017 to determine its ionic composition at predestinated points and to relate such properties to its physicochemical characteristics. Studies relating physicochemical properties and dissolved toxic ions in water could develop a body of knowledge that could enable detection and quantification of potential risk of ions such as heavy metals from natural water to aquatic ecosystem, animal and human health without actually involving aquatic organism, animal and human. Clean sterile plastic bottles were used for collecting surface water. A total of 30 sub-samples from five points at 300 m apart were sampled in the morning. Physicochemical properties were determined using standard methods and ionic composition of water was determined according methods of APHA. Results revealed that $Ca^{2+}$ had a mean $23.60 \pm 0.67$ mg/l and was the highest while $K^+$ with a mean $0.72 \pm 0.30$ was the least amongst major cations. Amongst the major anions $Cl^-$ had mean of $31.58 \pm 4.47$ mg/l while mean of $PO_4^{3-}$ was $1.42 \pm 0.13$ mg/l. The ionic balance calculate as % balance error showed high values for all sampling sites ranging from 30 to 39.42% indicating that there is massive input from anthropogenic activities. The computed relationships for selected heavy metals, cations and anions revealed that $R^2$ values were ranging between $\pm 0.012$ to 1 indicating some form of relationship existing. The water pH weakly correlated with dissolved cations and anions while moderate with pH only, due to the pH level (5.2–6.2). The cations and anions were more influenced by the water temperature than the heavy metals. Therefore, high temperature ranges of 31–32.4 °C will favour more dissolution of cations and anions in natural water. Cations showed stronger relationship with EC while only heavy metals showed no relationship with DO (Dissolved oxygen). Dissolved oxygen relationship with cations and anions was in the order; $K^+ > Mg^{2+} > Ca^{2+} > Na^+$ while anions was $SO_4^{2-} > NO_3^- > Cl^- > PO_4^{3-}$, respectively. Information here could be used to predict the effects of using this water for various purposes including water

for agricultural purposes, in the management of ion polluted waters, and also to inform on the mitigation process to be taken.

## INTRODUCTION

All waters in the environment contain dissolved salts existing in ionic forms. However, some species occur more frequently and at greater concentrations than others. The concentrations of dissolved salts in water are influenced by anthropogenic and natural factors such as industrial and domestic effluents and sewage, agricultural effluents, radioactive wastes, thermal pollution, oil pollution, topography, geology, and inputs through rainwater, water/rock interaction and climate (*Verla et al., 2018*; *Isiuku & Enyoh, 2019*). The occurrence of ions such as cations, heavy metals and anions in excess of natural load is of current concern not only to researchers, governmental and non-governmental organizations. The concern stems from their persistence in the environment and biopersistence when ingested. Therefore, causing damage to ecosystem and posing a serious health threat to the immediate population.

The distribution, solubility and mobility of ions in water are of importance to water as a media due to potential toxicity to man, plants and animals (*Enyoh et al., 2017*; *Isiuku & Enyoh, 2019*). The toxicity of dissolved ions such as heavy metals is due to their ability to bind to oxygen, nitrogen, and sulphur groups in proteins, resulting in alterations of enzymatic activity. Most organ/systems are affected by heavy metal toxicity; the most common include the hematopoietic, renal, and cardiovascular organs/systems (*Verla et al., 2019b*). Movement and chemical stability of ions in water are controlled by a complex series of biogeochemical processes that depend on physicochemical properties of the water such as pH, temperature, redox potential including adsorption, precipitation and ion exchange reactions in water (*Verla, Verla & Enyoh, 2017*; *Enyoh, Verla & Egejuru, 2018*). Generally, for most metals decreasing pH causes an increase in metal solubility in many forms except metals present in the form of oxyanions or amphoteric species. Studies have shown that metal solubility correlated positively with pH (*Enyoh, Verla & Egejuru, 2018*) and becomes limited at pH range of 5.5 to 6.0 while more distributed at temperature range of 15–35 °C (*Pérez-Esteban et al., 2013*; *Yang et al., 2006*; *Jing, He & Yang, 2007*; *Yuanxing et al., 2017*). Therefore, the distribution, solubility and toxicity of ions in water can be controlled by controlling the physicochemical properties of the water.

River systems, especially the ones flowing through urban cities, are greatly threatened by pollution. The Nworie River, which flows through Owerri, the capital city of Imo State in South-eastern Nigeria, is an example of such river system. Numerous studies have confirmed that the river is polluted by ions such as Cd, Ni, Fe, Hg, As, Co, Cu, Mn, $SO_4^{2-}$, $PO_4^{3-}$ and $NO_3^-$ (*Ukagwu et al., 2014*; *Verla et al., in press*) and without treatment could pose serious health issues. However, study focusing on the distribution of

its ionic composition and the relationship with its physicochemical properties is lacking. It is possible to characterize waters by performing a chemical analysis of their ionic composition. Such study reveals the nature of weathering and a variety of other natural and anthropogenic processes. Clearly, the precise chemical composition of the water will depend upon the types of rock and soils with which the water has been in contact and this can be used to characterize particular water by determining its chemical make-up and suggesting pollution problem mitigation strategy. Furthermore, studies focusing on the relationship between physicochemical parameters and ionic compositions of natural water are scarce, while laboratory studies have been well conducted. This kind of studies relating physicochemical properties and dissolved toxic ions in water could develop a body of knowledge that could enable detection and quantification of potential risk of ions such as heavy metals from natural water to aquatic ecosystems, animal and human health without actually involving an aquatic organism, animal or human.

## MATERIALS AND METHODS

### Study area

Nworie River is the study area (Fig. 1). The river originates from Mbiatolu LGA of Imo State between latitude 5°28N and 5°31N (Fig. 1). It passes through Owerri Municipal of Imo State and then empties into the Otamiri river at Nekede, Owerri West LGA, Imo State. The measured length of the river is approximately 9 kilometers. Photographs showing the decline state and anthropogenic activities of some points along Nworie River are shown in Figs. 2A–2D and Table 1 respectively.

### Collection of water sample

A total of 30 sub-samples were collected to make five composite (six subsamples per sampling location) samples. Samples were collected following a 'W' shaped design along the longitudinal course of the river using the grab technique (*Verla et al., in press*). From where the river enters the Owerri municipal to where it leaves. Geographically, the sampling points lies between the latitudes 05.52°N and longitude 07.03°E, NDW1 and NDW2 (Amakohia-Alvan), NDW3 (HolyGhost college), NDW4 (Wetheral) lies between 5.479°N and longitude 07.027°E, and NDW5 (river leaving the town) (Fig. 1). The water samples were collected during the dry season period. The points of sample collection were at least 300 m apart, done in morning against the water current. Clean plastic bottles were used for the collection.

### Analytical procedure of water sample

The electrical conductivity was measured using HANNA HI8733 EC METER in $\mu$S/cm, which was calibrated using KCl. The electrical conductivity (EC) probe was dipped into the water sample for 60 s and readings were recorded on the meter screen.

The pH was determined using JENWAY 3510 pH METER which was calibrated using buffer 4 and buffer 7 by dissolving one capsule each in 100 ml of distilled water respectively. The pH was determined by introducing the probe of the pH meter into water sample (collected from a large sample) and the reading on the meter screen was recorded.

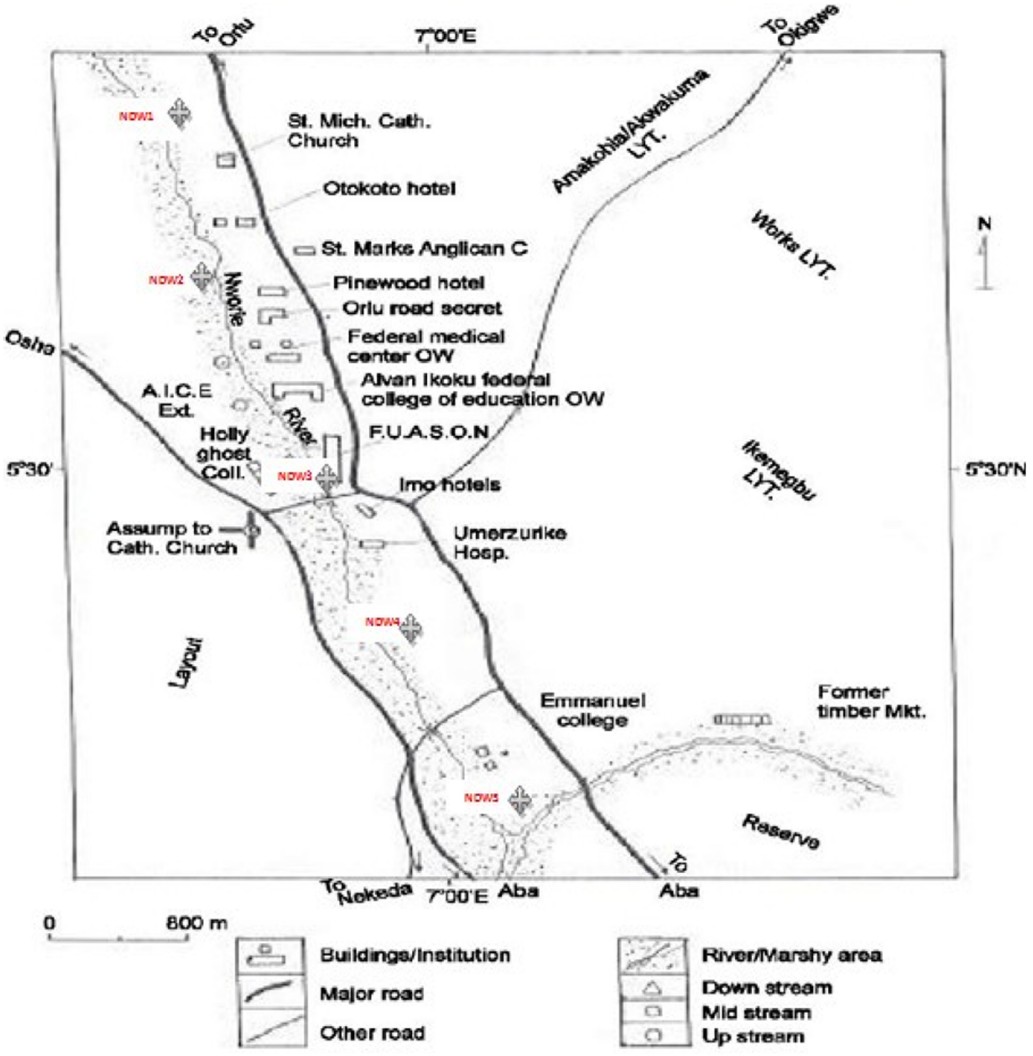

**Figure 1** Map showing Nworie River (*Chukwuma, Nwokedi & Noah, 2015* with slight modification).

Dissolved oxygen (DO) concentrations were determined with a Jenway 9071 DO analyzer by inserting the probe into the water samples.

TDS (Total Dissolved Solids) and TSS (Total Suspended Solid) (mg/L) was determined according method described by *Verla et al. (2018)* while phosphate ($PO_4^{3-}$), nitrate ($NO_3^{2-}$), chloride was determined according to American Public Health Association method 4110 (*APHA, 2005*).

## Determination of heavy metals and macro elements

The water sample was digested using aqua-regia ($HCl+HNO_3$ in ratio 3 to 1). one mL of the water sample was digested for 3 days in a test tube with 24 mL of aqua-regia (*Enyoh, Verla & Egejuru, 2018*; *Verla et al., 2018*; *Verla et al., 2019a*). The digested filtrates were used for the total metal quantification of Na, K, Mg, Ca, Mn Pb, Cd, Fe, Zn, and Cu using

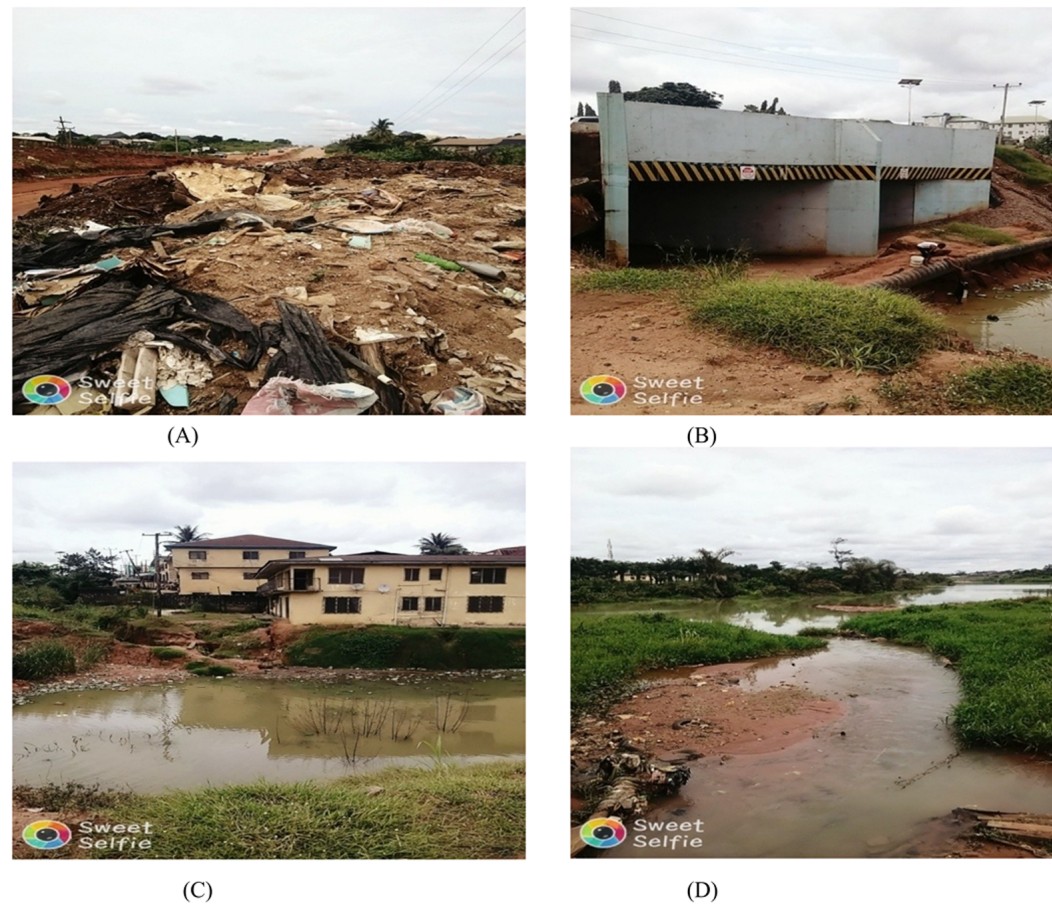

**Figure 2** **Photographs of some points along Nworie River (A–D).**

Atomic Absorption Spectrophotometer (Perkin Elmer AAnalyst 400) in mg/l (ppm). The characteristic wavelengths of metals determined were first set using the hollow cathode lamp, then digested filtrates samples was aspirated directly into the flame. To ensure accuracy of data, calibration of the equipment was done for each element using a standard sample prepared as a control with every set of samples. The instrumental parameters for particular metals that were analysed are presented in Table 2. The same procedure was done for all five composite samples.

## Data analysis

The data were analyzed for basic descriptive statistics such minimum, maximum, mean and standard deviation of triplicate analysis and level of significance was determined using Microsoft Excel and IBM SPSS statistics version 20. Correlation, Principal Component and linear regression analysis were conducted to establish relationship between physicochemical properties and ionic composition at 5% level of significance. P-values were considered significant when less than 0.05. To interpret the r-value for the correlation and linear relationship, the following classification was adopted, presented in Table 3. Hierarchical

**Table 1 The description and activities carried out within the study area.**

| Sampling points | Major Landmark | Human activity | Vegetation | Extent of Human destruction of Natural ecosystem |
|---|---|---|---|---|
| NDW1 | Fly-over bridge | Road expansion/construction, waste dumpsites, automechanic activities | Grassland and patches of dead grass. | [a] |
| NDW2 | Fly-over bridge | Road expansion/construction, waste dumpsites and automechanic activities | Grassland and patches of dead grass. | [a] |
| NDW3 | Bridge | Road expansion/construction, waste dumpsites and automechanic activities | Shrubs | [b] |
| NDW4 | Bridge, Mbari Kitchen restaurant and Umezurike Hospital | Road expansion/construction waste dumpsites, Dredging, sand mining and automechanic activities | Shrubs | [a] |
| NDW5 | Bridge and Emmanuel college | Road expansion/construction waste dumpsites and automechanic activities | Shrubs | [a] |

**Notes.**
[a] Very severe.
[b] Severe.

**Table 2 Optimal Instrumental parameters for AAS determination of the metals.**

| Metal symbols | Wavelength (nm) | Spectral Band Width (nm) | Flame gases | Time of measurement (secs) | Atomization flow rate (L/min) |
|---|---|---|---|---|---|
| Ca | 422.7 | 0.7 | Air-Acetylene | 4 | 1.2 |
| K | 766.5 | 0.7 | Air-Acetylene | 4 | 1.2 |
| Mg | 285.2 | 0.7 | Air-Acetylene | 4 | 1.1 |
| Na | 589.0 | 0.2 | Air-Acetylene | 4 | 1.2 |
| Pb | 283.3 | 0.7 | Air-Acetylene | 4 | 0.9 |
| Cu | 324.8 | 0.7 | Air-Acetylene | 4 | 0.9 |
| Fe | 248.3 | 0.2 | Air-Acetylene | 4 | 0.9 |
| Mn | 279.5 | 0.2 | Air-Acetylene | 4 | 0.9 |
| Zn | 213.9 | 0.7 | Air-Acetylene | 4 | 0.9 |
| Cd | 228.8 | 0.7 | Air-Acetylene | 4 | 0.9 |

Cluster Analysis (HCA) was used also to find similarities between physicochemical properties and ionic composition.

## RESULTS

### Surface water characteristics

The surface water characteristics are presented in Table 4. The results are compared with World Health Organization standards permissible limit (*WHO, 2007*). The mean values of temperature and total dissolved solids (TDS) exceeded the recommended limit. Mean

**Table 3  Classification system for the correlation and linear regression analysis.**

| $R^2$-value | Indication |
| --- | --- |
| −1 | A perfect negative linear relationship; dissimilar source of contamination |
| ≥ −0.70 | A strong negative linear relationship; dissimilar source of contamination |
| ≥ −0.50 | A moderate negative relationship; dissimilar source of contamination |
| ≥ −0.30 | A weak negative linear relationship; dissimilar source of contamination |
| 0 | No linear relationship |
| ≥ 0.30 | A weak positive linear relationship; similar source of contamination |
| ≥ 0.50 | A moderate positive relationship; similar source of contamination |
| ≥ 0.70 | A strong positive linear relationship; similar source of contamination |
| +1 | A perfect positive linear relationship; similar source of contamination |

**Table 4  Characteristics of Nworie River.**

| Parameters | NDW1 | NDW2 | NDW3 | NDW4 | NDW5 | WHO (2007) | Max | Min | Mean | SDV |
| --- | --- | --- | --- | --- | --- | --- | --- | --- | --- | --- |
| Temp (°C) | 31 | 32 | 32 | 32.4 | 32 | 20–30 | 32.4 | 31 | 31.88 | 0.55 |
| pH | 5.61 | 5.48 | 6.2 | 5.59 | 5.2 | 6.5–9.0 | 6.2 | 5.2 | 5.62 | 0.37 |
| EC (μS/cm) | 90 | 91 | 92 | 92 | 97 | 100 | 97 | 90 | 92.40 | 2.70 |
| DO (mg/L) | 2.78 | 2.98 | 2.85 | 3.89 | 2.27 | 4 | 3.89 | 2.27 | 2.95 | 0.59 |
| TDS (mg/L) | 123.94 | 123.84 | 143.78 | 122 | 123.98 | 250 | 143.78 | 122 | 127.51 | 9.13 |
| TSS (mg/L) | 87.74 | 86.64 | 96.74 | 96.43 | 87.44 | 50 | 96.74 | 86.64 | 90.99 | 5.12 |

**Notes.**
EC, Electrical conductivity; DO, Dissolved Oxygen; TDS, Total Dissolved Solids; TSS, Total Suspended solids.

pH (5.62) was below the acceptable range while other physicochemical parameters didn't exceed the permissible limit. However, the highest and lowest temperature was recorded at NDW4 and NDW1 respectively. The overlying water in NDW4 had the highest DO value of 3.89 mg/L and in NDW5 the lowest was recorded (2.27 mg/L). The standard DO expected according to WHO permissible limit (2007) is 4 mg/L, hence suggesting relatively poor water quality of Nworie River.

## Ionic composition and distribution

The results for the ionic compositions and percentage distributions are presented in Table 5 and Fig. 3. Only calcium, iron, nitrate, phosphate and chloride showed lower mean concentrations when compared to WHO permissible limits (Table 5), all others showed higher mean concentrations. The order of ionic composition was $Cl^-$ >$SO_4^{2-}$ >$Ca^{2+}$ >$Zn^{2+}$ >$Mg^{2+}$ >$NO_3^-$ >$PO_4^{3-}$ >$Na^+$ >$Mn^{2+}$ >others. Chloride, calcium and zinc showed the highest distributions for anions, cations and heavy metals respectively.

**Table 5  Ionic composition of water from Nworie in the dry season.**

| Ions | NDW1 | NDW2 | NDW3 | NDW4 | NDW5 | WHO (2007) | Max | Min | Mean | SDV |
|---|---|---|---|---|---|---|---|---|---|---|
| Major cations | | | | | | | | | | |
| $Na^+$ (mg/L) | 1.33 | 1.673 | 1.37 | 1.30 | 1.37 | N/A | 1.673 | 1.3 | 1.41 | 0.15 |
| $K^+$ (mg/L) | 0.89 | 0.791 | 0.898 | 0.189 | 0.819 | N/A | 0.898 | 0.189 | 0.72 | 0.30 |
| $Mg^{2+}$ (mg/L) | 2.23 | 2.78 | 2.27 | 2.22 | 1.13 | 0.5 | 2.78 | 1.13 | 2.13 | 0.60 |
| $Ca^{2+}$ (mg/L) | 23.2 | 24.6 | 23.8 | 23.5 | 22.92 | 70 | 24.6 | 22.92 | 23.60 | 0.67 |
| Heavy metals | | | | | | | | | | |
| Cu (mg/L) | 0.13 | 0.79 | 0.53 | 0.15 | 0.13 | 0.3 | 0.79 | 0.13 | 0.35 | 0.30 |
| Cd (mg/L) | 0.002 | 0.180 | 0.072 | 0.006 | 0.012 | 0.003 | 0.18 | 0.002 | 0.05 | 0.08 |
| Mn (mg/L) | 0.08 | 1.02 | 0.78 | 0.085 | 0.089 | 0.4 | 1.02 | 0.08 | 0.41 | 0.45 |
| Zn (mg/L) | 2.6 | 1.2 | 2.21 | 2.61 | 2.63 | <0.1 | 2.63 | 1.2 | 2.25 | 0.61 |
| Fe (mg/L) | 0.097 | 0.19 | 0.191 | 0.097 | 0.091 | 0.3 | 0.191 | 0.091 | 0.13 | 0.05 |
| Pb (mg/L) | 0.127 | 0.178 | 0.521 | 0.123 | 0.127 | 0.01 | 0.521 | 0.123 | 0.22 | 0.17 |
| Major anion | | | | | | | | | | |
| $NO_3^-$ (mg/L) | 1.96 | 1.91 | 1.92 | 1.91 | 1.916 | 10 | 1.96 | 1.91 | 1.92 | 0.02 |
| $SO_4^{2-}$ (mg/L) | 24.8 | 23.4 | 24.8 | 23.98 | 24.8 | 250 | 24.8 | 23.4 | 24.36 | 0.64 |
| $PO_4^{3-}$ (mg/L) | 1.37 | 1.65 | 1.36 | 1.34 | 1.37 | 5 | 1.65 | 1.34 | 1.42 | 0.13 |
| $Cl^-$ (mg/L) | 24.15 | 32.24 | 36.19 | 33.16 | 32.18 | 250 | 36.19 | 24.15 | 31.58 | 4.47 |

**Notes.**

WHO,  World Health Organization.

## Ion balancing of River

When a water quality sample has been analysed for the major ionic species, one of the most important validation tests can be conducted: the cation-anion balance (*HPTM, 1999*). The principle of electroneutrality requires that the sum of the positive ions (cations) must equal the sum of the negative ions (anions). Thus the error in a cation-anion balance can be written as (1):

$$\%Balance\ error = \frac{\sum Cations\ in\ the\ river - \sum Anions\ in\ the\ river}{\sum Cations\ in\ the\ river + \sum anions\ in\ the\ river}. \tag{1}$$

The computed cation-anion balance is presented in Fig. 4. For surface water, the % error should be less than 10. Here, a balance error ranging from 30 to 39.42% is recorded for the different sampling point of the river. The high balance errors suggest that major ionic concentrations distributed in Nworie River are not equal (i.e major cations $\neq$ major anions).

## Correlation analysis and Principal component analysis of various ions

The result for the Pearson's correlation analysis of various ions in Nworie river is presented in Table 6. This model has been used well by many researchers in determining contamination sources for pollutants in the environment (*Enyoh, Verla & Egejuru, 2018*; *Duru, Okoro & Enyoh, 2017*; *Verla, Verla & Enyoh, 2017*; *Ibe, Isiukwu & Enyoh, 2017*). The value of *r* is always between +1 and −1. Most ions exhibit strong relationships, suggesting that their presence in the water is from similar anthropogenic source(s). Strong relationship was exhibited by some of the ions. Cations exhibited strong relationship amongst them. Na

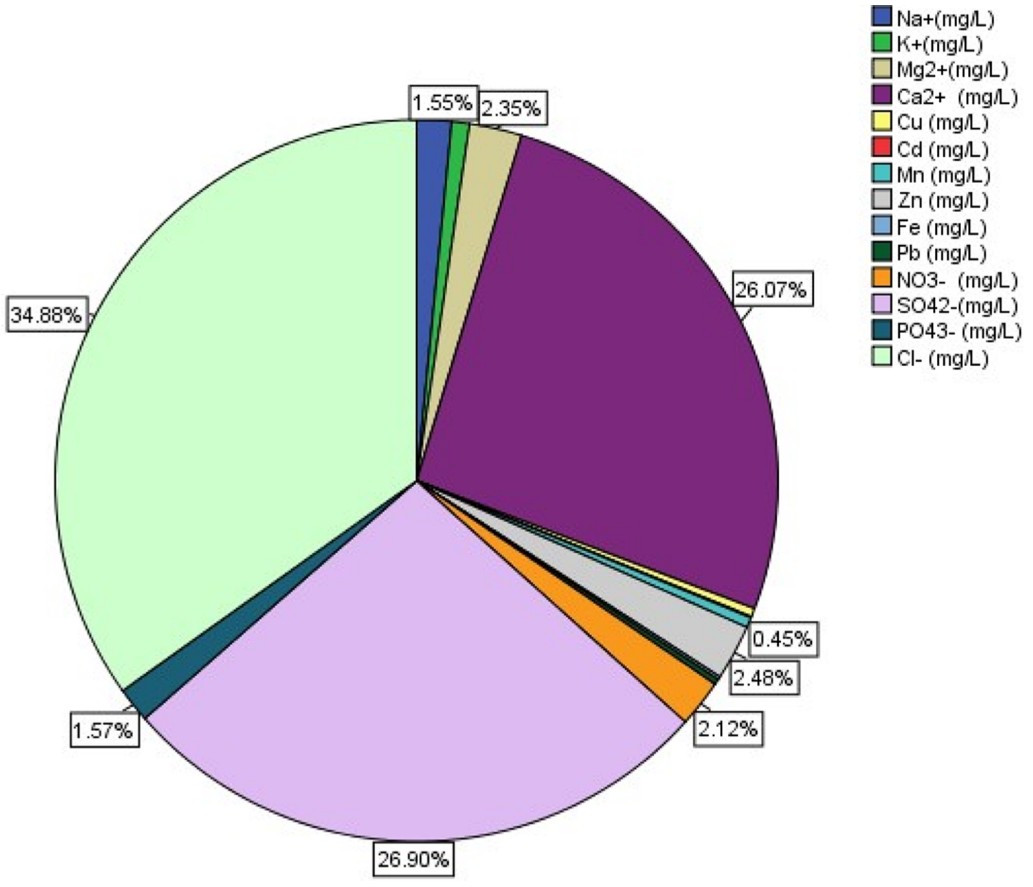

**Figure 3   Ionic distribution in the Nworie River.**

further showed strong relationship with some heavy metals (Cu, Cd, Mn and Fe) and anion ($PO_4^{3-}$). Similar relationships were also exhibited $Mg^{2+}$ and $Ca^{2+}$. Strong relationship was exhibited between heavy metals amongst them and with $Cl^-$ and $PO_4^{3-}$, except for Zn. Most anions showed negative relationships except for $NO_3^-$ / $SO_4^{2-}$ (0.54). To determine the precise contamination source(s) of ions as predicted by the correlation analysis; we conducted a principal component analysis following standard procedures (*Dragović, Mihailović & Gajić, 2008*; *Franco-Uría et al., 2009*). We used the varimax rotation with Kaiser Normalization because it better explained the possible groups or sources that influence the soil system and maximise the sum of the variance of the factor coefficients (*Gotelli & Ellinson, 2013*). The factor loadings for heavy metals in the river water are shown in Table 7. Three components were extracted based on eigen value >1, best described the sources.

## Relationship between water characteristics and dissolved ions

In order to establish a relationship between the recorded physicochemical properties and ionic composition of Nworie river, multiple linear regression analysis was conducted. Linear regression is a statistical method used to assess a possible linear association between

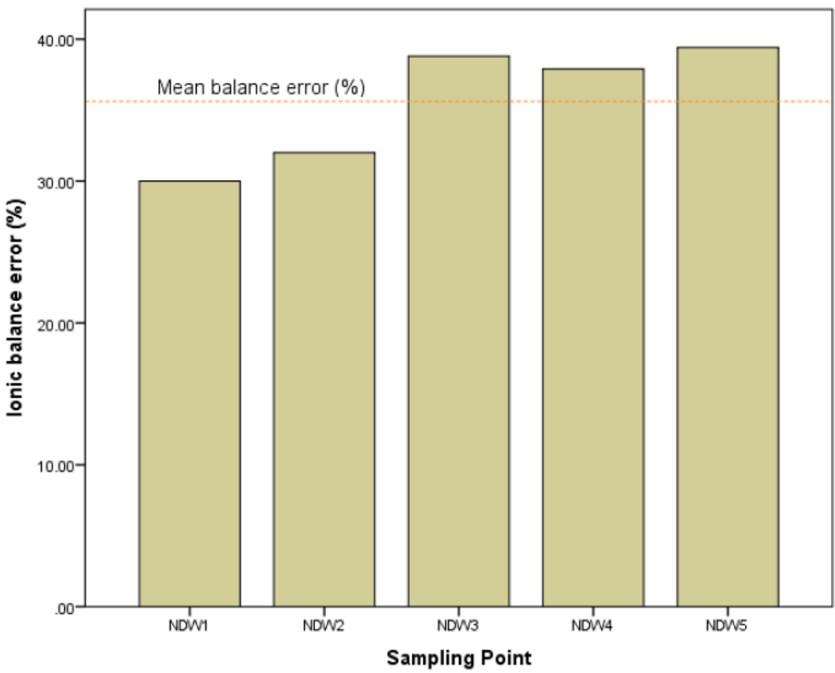

**Figure 4   Computed ion balance error at the different sampling points of the Nworie River.**

two continuous variables. The analysis gives out a regression coefficient ($R^2$ value) (*Enyoh, Verla & Egejuru, 2018*). $R^2$ value reveals the extent of influence of river physicochemical properties on the distribution of ions in the river on the basis of closeness to either –1 or +1 to indicate a strong enough linear relationship. The computed relationship for cations, anions and heavy metals are presented in Figs. 5–7. All ions showed positive relationships at varying level with the water physicochemical characteristics.

### Hierarchical Cluster analysis

The result for Hierarchical cluster analysis (HCA) is presented in Fig. 8, most the ions showed similarities with measured physicochemical parameters. Only TDS and EC showed some dissimilarity with iron and sulphate.

## DISCUSSION

### Surface water characteristics

Only mean values of temperature and total dissolved solids (TDS) exceeded the recommended limit (Table 2). The characteristics of the river are a reflection of the sampling period. High temperature obtained wasn't surprising due sampling season (dry season). Low values of DO have been reported earlier in other studies (*Manila & Frank, 2009*; *Duru & Nwanekwu, 2012*; *Okoro, Uzoukwu & Ademe, 2016*; *Verla et al., in press*; *Verla et al., 2019a*). These studies related the low DO to human activity, causing enrichment of the surface water with high organic content. Electrical conductivity recorded in this study fell below the acceptable limit (100 μS/cm) set by *WHO (2007)*. The conductivity depends

Verla et al. (2020), *PeerJ Analytical Chemistry*, DOI 10.7717/peerj-achem.5

**Table 6  Correlation matrix of various ions in Nworie river.**

| | Na$^+$ | K$^+$ | Mg$^{2+}$ | Ca$^{2+}$ | Cu | Cd | Mn | Zn | Fe | Pb | NO$_3^-$ | SO$_4^{2-}$ | PO$_4^{3-}$ | Cl$^-$ |
|---|---|---|---|---|---|---|---|---|---|---|---|---|---|---|
| Na$^+$ | 1 | | | | | | | | | | | | | |
| K$^+$ | 0.291 | 1 | | | | | | | | | | | | |
| Mg$^{2+}$ | **0.515** | −0.081 | 1 | | | | | | | | | | | |
| Ca$^{2+}$ | **0.834** | 0.032 | **0.838** | 1 | | | | | | | | | | |
| Cu | **0.864** | 0.306 | **0.675** | **0.942** | 1 | | | | | | | | | |
| Cd | **0.953** | 0.272 | **0.638** | **0.938** | **0.975** | 1 | | | | | | | | |
| Mn | **0.805** | 0.357 | **0.643** | **0.907** | **0.992** | **0.941** | 1 | | | | | | | |
| Zn | **−0.968** | −0.240 | **−0.676** | **−0.942** | **−0.951** | **−0.993** | **−0.907** | 1 | | | | | | |
| Fe | **0.671** | 0.371 | **0.635** | **0.850** | **0.949** | **0.856** | **0.979** | **−0.808** | 1 | | | | | |
| Pb | −0.012 | 0.374 | 0.215 | 0.289 | 0.462 | 0.259 | **0.567** | −0.167 | **0.715** | 1 | | | | |
| NO$_3^-$ | −0.359 | 0.439 | 0.013 | −0.411 | −0.405 | −0.424 | −0.386 | 0.377 | −0.338 | −0.130 | 1 | | | |
| SO$_4^{2-}$ | **−0.726** | 0.431 | **−0.640** | **−0.789** | **−0.596** | **−0.700** | **−0.502** | **0.747** | −0.371 | 0.288 | 0.544 | 1 | | |
| PO$_4^{3-}$ | **0.989** | 0.223 | **0.568** | **0.831** | **0.818** | **0.923** | **0.745** | **−0.954** | **0.602** | −0.119 | −0.298 | **−0.780** | 1 | |
| Cl$^-$ | 0.143 | −0.220 | −0.012 | 0.345 | 0.429 | 0.334 | 0.475 | −0.245 | **0.524** | **0.598** | **−0.858** | 0.193 | 0.039 | 1 |

**Notes.**
*Bold numbers are significant at 5%.

**Table 7  Principal Component Matrix using varimax rotation of ions in the river water.**

| Parameter | Principal component | | |
|---|---|---|---|
| | PC1 | PC2 | PC3 |
| $Cd^{2+}$ | .990 | – | – |
| $Cu^{2+}$ | .985 | .166 | – |
| $Zn^{2+}$ | −.978 | .119 | −.147 |
| $Ca^{2+}$ | .975 | – | – |
| $Mn^{2+}$ | .958 | .285 | – |
| $Na^+$ | .901 | −.223 | .223 |
| $Fe^{2+}$ | .894 | .427 | – |
| $PO_4^{2-}$ | .873 | −.343 | .264 |
| $SO_4^{2-}$ | −.726 | .647 | .215 |
| $Mg^{2+}$ | .711 | −.158 | .161 |
| $Pb^{2+}$ | .329 | .904 | −.160 |
| $NO_3^-$ | −.506 | – | .814 |
| $Cl^-$ | .406 | .404 | −.804 |
| $K^+$ | .173 | .579 | .718 |
| **Eigen value** | 8.739 | 2.252 | 2.074 |
| **% Variance** | 62.424 | 16.088 | 14.812 |
| **% Cumulative** | 62.424 | 78.512 | 93.324 |

on water temperature and is the measure of water capability to pass electric flow. High temperature may cause high EC. The conductive ions may come from dissolved salts and inorganic materials in the River.

## Ionic composition and distribution

Dissolved salt often exists as ions in solution. The ions can be positive or negative. Ions with a positive charge are called "cations" while negative charge is called an "anion". Going by this definition, heavy metals are cations because they are positively charged. However in this study we classified the ions as major cations, heavy metals (often categorized as secondary constituents) and anions. The major cations studied were sodium, potassium, magnesium and calcium; heavy metals were copper, cadmium, manganese, zinc, iron and lead; while anions were nitrate, sulphate, phosphate and chloride. The results for the ionic compositions are presented in Table 4. The major cationic and anionic concentrations were low generally when compared to WHO permissible limit (2007) standard except for magnesium. The magnesium concentration ranges from 1.13 mg/L to 2.78 mg/L as opposed to the WHO permissible limit (2007) standard of 0.5 mg/L. Calcium ($Ca^{2+}$) and Magnesium ($Mg^{2+}$) ions are both common in natural waters and both are essential elements for all organisms (*HPTM, 1999*). They are responsible for the hardness of natural waters when combined with dissolved materials in the water. WHO reported that hard water has no known adverse health effect (*Sengupta, 2013*) and could provide an important supplementary contribution to total calcium and magnesium intake (*Galan et al., 2002*). However, prolong consumption of water with high magnesium can cause hypermagnesemia

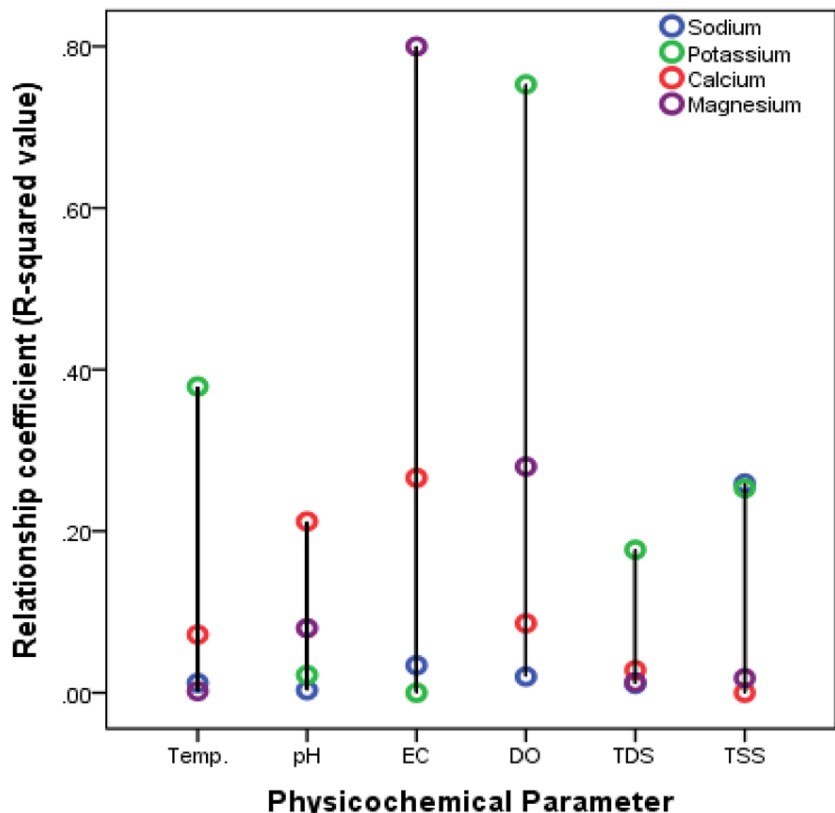

**Figure 5** **Relationship between cations and water characteristics of the Nworie River.**

if the consumer have significantly decreased ability to excrete magnesium (*Chandra et al., 2013*).

All heavy metals except for Fe had mean values exceeding the permissible limit set by WHO (Table 5), which is indicative that the water is polluted by these metals. The concentration of Pb ranged from 0.127 to 0.521 mg/l, Fe ranged from 0.091 to 0.191, Cd ranged from 0.002 to 0.180 mg/l, Cu ranged from 0.13 to 0.79 mg/l, manganese ranged from 0.08 to 1.02 and zinc ranged from 1.2 to 2.63 with mean values of 0.22, 0.13, 0.05, 0.35, 0.41 and 2.25 mg/l respectively. The solubility of trace metals in surface waters is predominantly controlled by the water pH, the type and concentration of ligands on which the metal could adsorb, and the oxidation state of the mineral components and the redox environment of the system. Ingestion of metals such as lead (Pb) and cadmium (Cd) may pose great risks to human health by interfering with essential nutrients in the body, possibly causing small increases in blood pressure and damaging the kidney. In addition, they can equally affect aquatic fauna and flora (*Isiuku & Enyoh, 2019*).

The percentages of ions are presented in Fig. 3. This distribution is showing the abundance of major ions in Nworie river during dry season. Metal ions and anions highly distributed were calcium (26.07%), sulphate (26.90%), chlorine (34.88%), while sodium, potassium, nitrate, phosphate, zinc were within the range of 1–3%. Other heavy metal ions
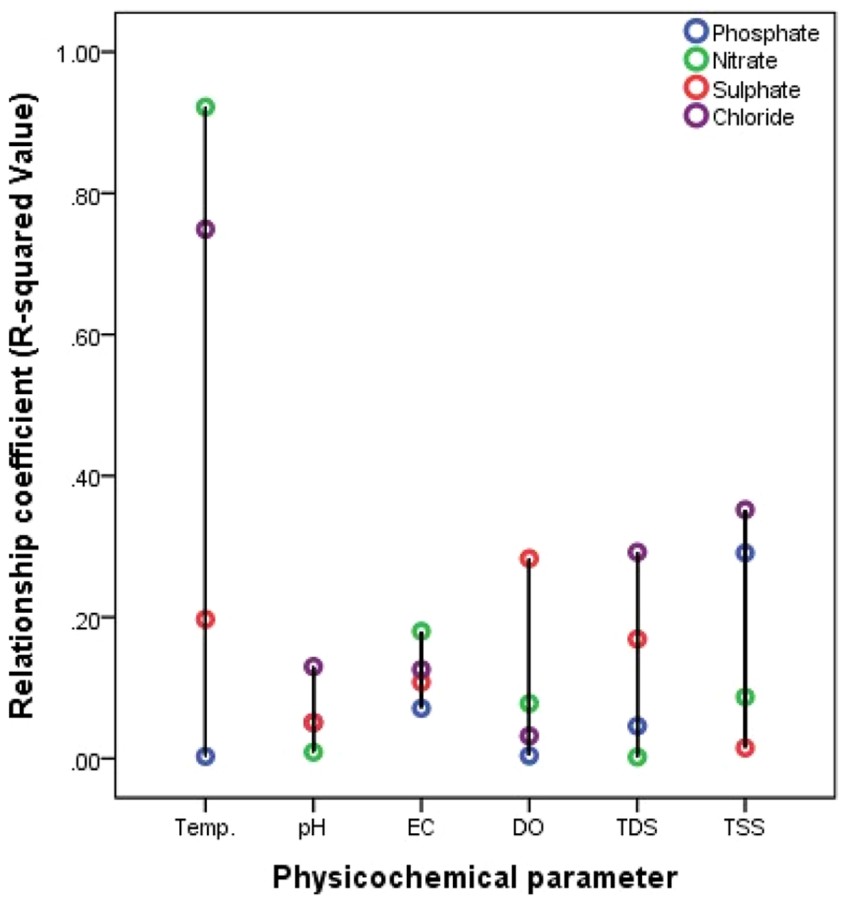

**Figure 6** **Relationship between anions and water characteristics of the Nworie River.**

showed distribution of 0%. These suggest that the presence of heavy metals in Nworie river during dry season is low in the total composition of ions in the River. The low distribution could probably due to the metals forming complexes with organic materials in the water or they could be in abundance in other forms which can be accessed through chemical speciation (*Verla et al., 2019c*).

## Ion balancing of River

The ionic balancing of the river is high (mean balance error of 35.62%) indicating variable dissolution of ions in the river (Fig. 4). The obvious reason could be due to pollution by anthropogenic means through dumping of waste and the river finding itself at the receiving end of an effluent discharge from domestic and industrial sources. Fertilizers from farmlands along Nworie river finds their way into the river through surface run-off and increases the anionic concentrations such as $PO_4^{3-}$ and $NO_3^-$ in the river while major cations might undergo complexometric reactions reducing their concentrations in the water. Anthropogenic activities have been reported to significantly alter ionic concentration in water (*Manila & Frank, 2009*; *Enyoh, Verla & Egejuru, 2018*; *Verla et al., 2018*; *Isiuku & Enyoh, 2019*).
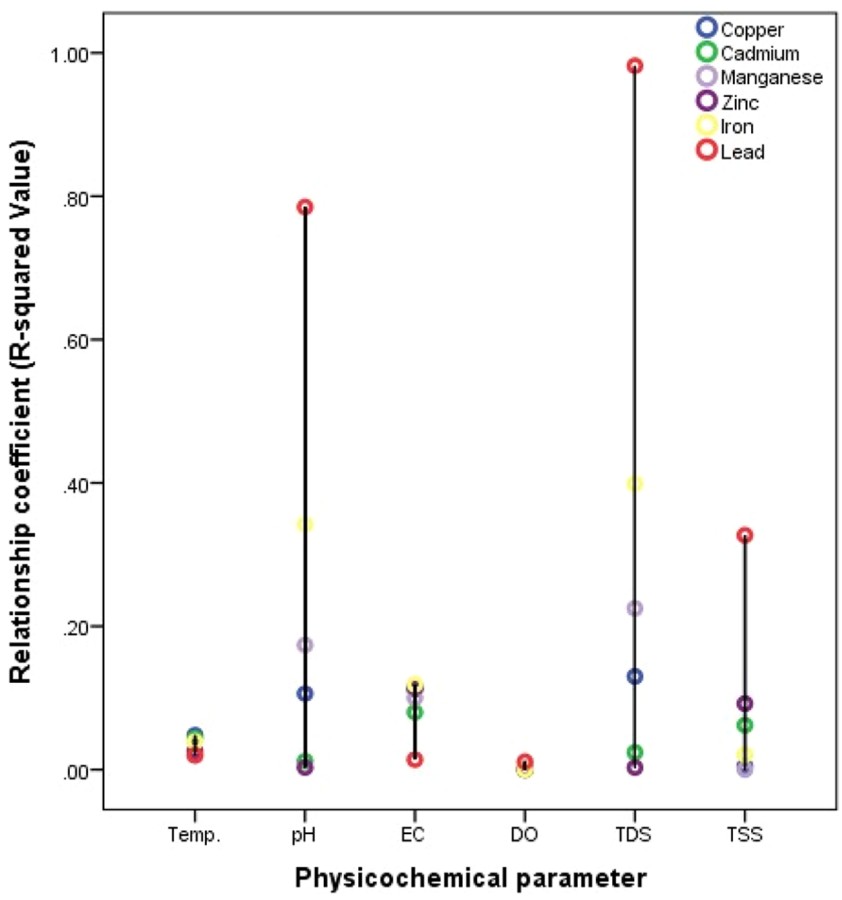

**Figure 7** **Relationship between heavy metals and water characteristics of the Nworie River.**

## Correlation analysis and Principal component analysis of various ions

The result for the Pearson's correlation analysis of various ions in Nworie river showed many ions showing strong relationships suggesting that their presence in the water is from similar anthropogenic source(s). Some association exhibited by the ions has observed in other water studies (*Enyoh, Verla & Egejuru, 2018*; *Duru, Okoro & Enyoh, 2017*; *Verla, Verla & Enyoh, 2017*; *Ibe, Isiukwu & Enyoh, 2017*). Precise contamination source of ions was determined by PCA. Three components were extracted based on the eigen value greater than 1. The three components cumulatively explained 93.324% of the total variance and generally indicated an anthropogenic source(s) of the studied ions in the river water (Table 7). According to *Liu, Lin & Kuo (2003)*, components loadings values of >0.75, 0.75–0.50, and 0.50–0.30 were classified as "strong", "moderate", and "weak respectively. The PC1 explained 62.424% of total variance and was found to be strongly and positively correlated to $Cd^{2+}$, $Cu^{2+}$, $Ca^{2+}$, $Mn^{2+}$, $Na^+$, $Fe^{2+}$, $PO_4^{2-}$ and $Mg^{2+}$ (0.71–0.99). This relates to the artisanal activities, metal processing works in the area and agricultural activities; PC2 explained 16.088% of total variance and showed moderate to strong positive factor loadings for $SO_4^{2-}$ and Pb (0.65–0.90) which also indicate industrial source, activities of

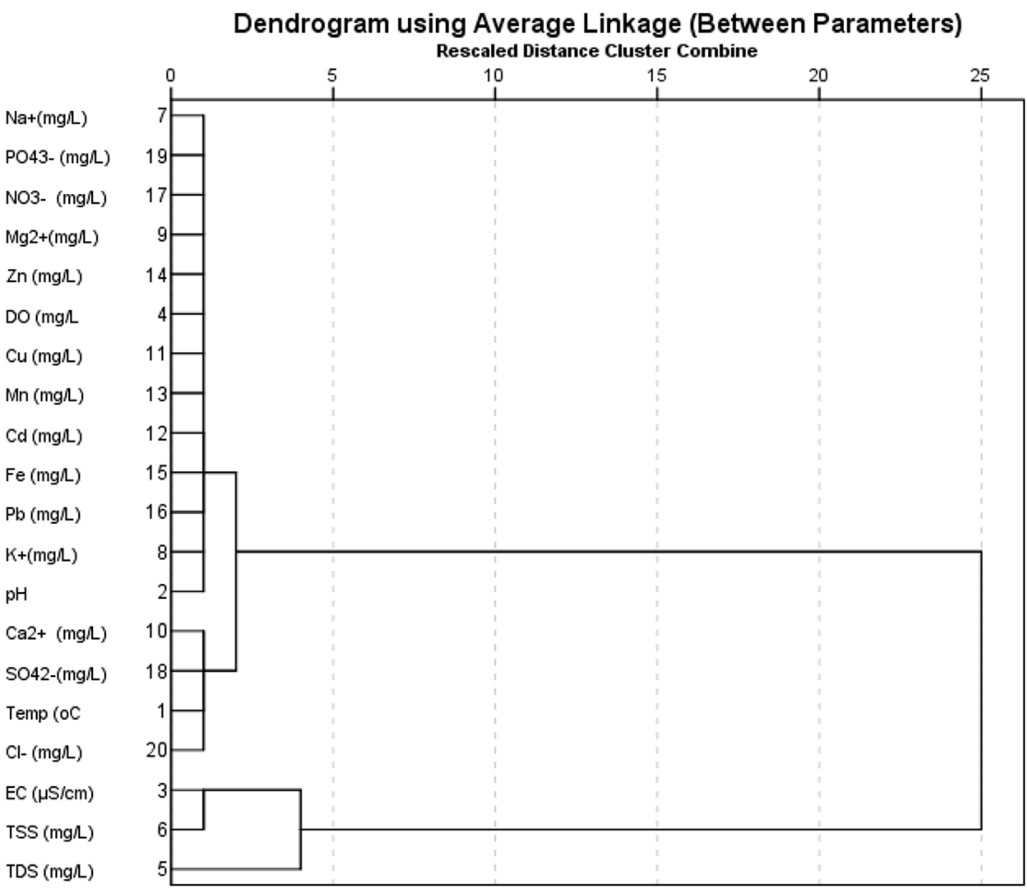

**Figure 8** **Dendrogram of physicochemical and ionic properties.**

scrap metal dealers, recycling of metals as well as lead–acid accumulators, vehicle emissions in urban areas and agricultural activities in the study area. PC3 explained 14.812% of total variance and showed strong positive factor loadings for $NO_3^-$ and $K^+$ (0.72–0.81) which could be attributed to atmospheric deposition and agricultural activities involving the use of fertilizers.

## Relationship between water characteristics and dissolved ions

The computed relationship for cations, anions and heavy metals are presented in Figs. 5–7.

### *Temperature*

Different researches have reported different result for temperature effects on metal distribution in water. most studies have been conducted under laboratory conditions. *Lourino-Cabana et al. (2014)* and *Green-Ruiz, Rodriguez-Tirado & Gomez-Gil (2008)* reported that the distribution of metals changed with temperature variation while also some research reported that the influence of temperature on metal distribution was not evident (*Aston et al., 2010*; *Biesuz et al., 1998*; *Zhang, Lee & Pan, 2013*). Another studies by *Echeverrıa et al. (2003)* and *Echeverría et al. (2005)* found that increased temperature resulted in a higher maximum sorption of metals by minerals. In the current study, the

temperature relationship was measured under ambient field/natural conditions. Our results showed that temperature relationship was weakly positive especially with heavy metals distribution with $R^2$- value <0.1 (Fig. 5). We also observed a strong positive relationship existed with anions such as $NO_3^-$ (0.922) and $Cl^-$ (0.749) (Fig. 6) while only $K^+$ (0.379) was more significant for major cations (Fig. 5). *Haiyan et al. (2013)* reported that at temperature range of 4–25 °C, there is a weak temperature-dependence of metal distributed. However in this study high temperature ranges of 31–32.4 °C were recorded (Table 4), which fell between the temperature ranges of 15—35 °C reported by *Yuanxing et al. (2017)*, who observed that distribution rates of ions such Zn, Cu, Pb, Cr, and Cd were greater in high temperatures than at low temperature.

## pH

Different studies have shown that heavy metals are generally associated with contaminated water (*Borma, Ehrlich & Barbosa, 2003*; *Lors, Tiffreau & Laboudigue, 2004*). *Yuanxing et al. (2017)* explains that when pH decreases (<5), the competition between $H^+$ and the dissolved metals for ligands (e.g., $OH^-$, $CO_3^{2-}$, $SO_4^{2-}$, $Cl^-$, $S_2^-$ and phosphates) becomes more and more significant thus increasing the mobility of heavy metals (*Borma, Ehrlich & Barbosa, 2003*). This explanation had been supported by many researchers (*Butler, 2009*; *Watmough, Eimers & Dillon, 2007*; *Pérez-Esteban et al., 2013*; *Yang et al., 2006*; *Jing, He & Yang, 2007*). However, the low pH instances are very rare in natural environment (*Yuanxing et al., 2017*) where pH values cannot be easily controlled. A critical look at Figs. 5–7 for effects of pH, it would be observed that the relationship was more pronounced with heavy metals especially with Pb (0.785) and Fe (0.342), then major cations followed the order; $Ca^{2+}$ (0.212) >$Mg^{2+}$ (0.080) while generally low with anions (<0.2). Most of these anions serves as ligands for complex formation with metals and thus could be reason for the poor relationship. *Yuanxing et al. (2017)* in a controlled experiments reported that at low-pH (pH = 4), the distribution of some metals especially Zn in solution was much larger than at high-pH or mid-pH. The pH recorded in our study was 5.2 to 6.2, which could possibly be the reason for general low distribution hence suggesting low pH-dependence.

## EC

The effects of EC on ionic distributions are presented in Figs. 5–7. The EC showed positive correlations with the ions studied and followed the order; cations (0.28) >anions (0.12) >heavy metals (0.09) respectively. These suggest that EC-dependence for ion distribution was more significant with cations. This however isn't surprising since EC measures electrolytes in water and majorly controlled by cations (especially $Ca^{2+}$, $Mg^{2+}$ and $Na^+$) and anions which are more soluble than heavy metals.

## DO

The effects of DO on the ionic compositions are generally low but positive. However, no effect was recorded for heavy metals in the river with coefficient of zero (0) (Fig. 5). *Haiyan et al. (2013)* in a controlled experiment reported that the release of some heavy metals such as Cu, Pb, and Cr was much faster at DO >5 mg/L than DO <5 mg/L. Therefore, no effect recorded here on the distribution of studied heavy metals could be due to the low

DO (2.78 to 3.89 mg/L) of the river at the time of study. However, the effect of DO on the distribution of major cations followed the order; $K^+$ (0.753) $>Mg^{2+}$ (0.280) $>Ca^{2+}$ (0.086) $>Na^+$ (0.020) (Fig. 5) while anions followed the order $SO_4^{2-}$ (0.283) $>NO_3^-$ (0.078) $>Cl^-$ (0.032) $>PO_4^{3-}$ (0.004) respectively (Fig. 6).

### TDS and TSS

The effects of TDS and TSS on ionic compositions are presented in Figs. 5–7. The TDS-dependence followed the order; Heavy metals (0.327) >anions (0.057) >cations (0.057), while the TSS-dependence followed the order; anions (0.186) >cations (0.133) >heavy metals (0.100). This could be from the weight of the ions. Heavy metals are heavier (with atomic weights between 63.546 and 200.590) and less dissolved in water when compared to anions and cations.

### Hierarchical Cluster analysis

Hierarchical cluster analysis (HCA) based on Square Euclidian Distance (SED) was carried out to show the similarities and/or dissimilarities between the physicochemical characteristic data and ionic concentrations in the river water. The SED was adopted since it is based on the Euclidian Distance between two observations, which is the square root of the sum of squared distances, thereby increases the importance of large distances, while weakening the importance of small distances. The results obtained from this analysis are presented as dendrogram shown in Fig. 8. A dendrogram is a branch diagram that represents the level of relationships or similarity among parameters arranged like the branches of a tree (*Ibe et al., 2019*). From Fig. 8, based on the rescaled distance of cluster combined, it can be seen that most the ions showed similarities with measured physicochemical parameters. Only TDS and EC showed some dissimilarity with iron and sulphate. The similarities exhibited by most parameters further highlighted the relationships existing between the physicochemical and ionic parameters.

## CONCLUSION

The study has shown that the physicochemical properties of river water from Nworie river influences ionic composition under ambient natural/field conditions. The presence of ions dissolved in the river water as revealed by the PCA and HCA was from anthropogenic activities including atmospheric deposition, application of fertilizer on nearby farming, artisanal and automechanic activities. The ions showed varying relationship with the water characteristics. The water pH weakly correlated with dissolved cations and anions while moderate with pH only, due to the pH level (5.2–6.2). The cations and anions were more influenced by the water temperature than the heavy metals. Therefore, high temperature ranges of 31–32.4 °C will favour more dissolution of cations and anions in natural water. Cations showed stronger relationship with EC while only heavy metals showed no relationship with DO. Dissolved oxygen relationship with cations and anions was in the order; $K^+$ $>Mg^{2+}$ $>Ca^{2+}$ $>Na^+$ while anions was $SO_4^{2-}>NO_3^-$ $>Cl^-$ $>PO_4^{3-}$ respectively. The TDS and TSS had more influence on heavy metals and anions respectively.

All the factors studied in this study merit attention in the management of metals and anions polluted waters and also inform on the mitigation process to be taken.

### Funding

This research was funded by grants from Group Research in Analytical Chemistry, Environment and Climate Change (GRACE&CC/2019/H2O/08). The funders had no role in study design, data collection and analysis, decision to publish, or preparation of the manuscript.

### Grant Disclosures

The following grant information was disclosed by the authors:
Group Research in Analytical Chemistry, Environment and Climate Change (GRACE&CC/2019/H2O/08).

### Competing Interests

The authors declare there are no competing interests.

### Author Contributions

- Evelyn Ngozi Verla and Andrew Wirnkor Verla conceived and designed the experiments, performed the experiments, authored or reviewed drafts of the paper, and approved the final draft.
- Christian Ebere Enyoh performed the experiments, analyzed the data, prepared figures and/or tables, authored or reviewed drafts of the paper, and approved the final draft.

### Data Availability

The raw data shows the characteristics and ionic composition of the river and is available in the Supplemental Files.

### Supplemental Information

Supplemental information for this article can be found online at http://dx.doi.org/10.7717/peerj-achem.5#supplemental-information.

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
