# Peer review of "Finding a relationship between physicochemical characteristics and ionic composition of River Nworie, Imo State, Nigeria"

_PeerJ Analytical Chemistry, doi:10.7717/peerj-achem.5_

## Round 0.1 · original submission · Major Revisions

· Academic Editor

Major Revisions

Both the reviewers expressed significant concern regarding the writing style and it requires major revision of English used. Also, the limited amount of data presented is not enough to prove the validity of the research. Hence, more data is required to validate the graphs given. Presenting simple statistics might not be enough and require the presentation of original data which lead to the statistics presented (e.g. UV/Vis spectrum). Once the manuscript is thoroughly revised, it can be considered for publication, pending additional reviews.

Reviewer 1 ·

Basic reporting

The paper is difficult to read and several aspects has to be reconsidered, from the experimental plan to the data interpretation.

References can be improved by introducing more recent papers.

Experimental data are poor. No data regarding the methods and data quality.

Figures with low resolution and not very conclusive for the study.

English language has to be well improved.

Experimental design

It is a classical monitoring study of water quality.

The manuscript has a poor experimental plan. Less samples - only five - were considered for the study. No relevant conclusion can be draw.

Methods have been poor described.

Validity of the findings

There are no novelty in this study.

The data do not support a comprehensive study for establishing a relation between the physicochemical parameters and the ionic composition of water.

Additional comments

The title of manuscript in not really consistent with the text. In my opinion the paper is only a simple monitoring study of water quality in the River Nworie, Nigeria. It is a classical study analyzing specific quality indicators of water quality by standard methods. What is the novelty of the work?
The paper has many gaps and I cannot recommended for publication in its present form. The manuscript is difficult to read.
- The abstract has to be reconsidered, it has to highlight the main findings regarding the relationship between physicochemical characteristics and ionic composition of the Nworie River waters, as is stated in the title.
- The study was performed during the dry season period only on 5 samples. Why only 5 samples were considered? Normally, a more comprehensive study has to be done to draw real conclusions. Therefore, the experimental plan has to be reconsidered.
- A detailed description of the anthropogenic activities along the river has to be performed.
- Data on materials and methods used is very poor.
- Figures has low resolution.

Reviewer 2 ·

Basic reporting

I strongly recommend the authors to edit the language errors in the current version of the manuscript. This will help the readers to follow and understand the observations and discussions presented in this article.
Please double-check the figure numbers in the main body, figure captions as well as in the actual figure are consistent and are accurate. For instance, in line#146 it is indicated that the “percentage of ions are presented in Figure 2” which is not correct.

Experimental design

Although the authors had emphasized the importance of studying the relationship between physicochemical properties and chemical properties, the focus of the current study is not stated explicitly in the introduction of the manuscript.

Validity of the findings

In the manuscript, I don’t find any data for UV-Vis absorbance as well as other measurements based on which the relationship analysis is done. It is very important to show such experimental results that allow the readers to understand the measurement and estimation procedures followed by the authors. Due to the lack of any such experimental data in the manuscript, I find it difficult to follow the discussion and verify the conclusions derived through this study.

Additional comments

In tables 2 and 3, I would suggest using “WHO permissible limit (2007)” instead of just “WHO (2007)”

Annotated reviews are not available for download in order to protect the identity of reviewers who chose to remain anonymous.

---

## Round 0.2 · accepted · Accept

· Academic Editor

Accept

Thanks for considering the reviewers' suggestions and modifying the manuscript accordingly. I am glad to note that the reviewer found that the manuscript is modified sufficiently to warrant publication. Hence, we are pleased to accept the manuscript for publishing in PeerJ Analytical Chemistry.

Reviewer 2 ·

Basic reporting

No comment

Experimental design

No comment

Validity of the findings

No comment